# Synthetic modified vaccinia Ankara vaccines confer cross-reactive and protective immunity against mpox virus

Flavia Chiuppesi [1✉], John A. Zaia[2], Miguel-Angel Gutierrez-Franco[1], Sandra Ortega-Francisco [1], Minh Ly[1], Mindy Kha[1], Taehyun Kim[1], Shannon Dempsey[1], Swagata Kar[3], Alba Grifoni[4], Alessandro Sette [4], Felix Wussow [1] & Don J. Diamond [1]

## Abstract

**Background** Although the mpox global health emergency caused by mpox virus (MPXV) clade IIb.1 has ended, mpox cases are still reported due to low vaccination coverage and waning immunity. COH04S1 is a clinically evaluated, multiantigen COVID-19 vaccine candidate built on a fully synthetic platform of the highly attenuated modified vaccinia Ankara (MVA) vector, representing the only FDA-approved smallpox/mpox vaccine JYNNEOS. Given the potential threat of MPXV resurgence and need for vaccine alternatives, we aimed to assess the capacity COH04S1 and its synthetic MVA (sMVA) backbone to confer MPXV-specific immunity.

**Methods** We evaluated orthopoxvirus-specific and MPXV cross-reactive immune responses in samples collected during a Phase 1 clinical trial of COH04S1 and in non-human primates (NHP) vaccinated with COH04S1 or its sMVA backbone. MPXV cross-reactive immune responses in COH04S1-vaccinated healthy adults were compared to responses measured in healthy subjects vaccinated with JYNNEOS. Additionally, we evaluated the protective efficacy of COH04S1 and sMVA against mpox in mpox-susceptible CAST/EiJ mice.

**Results** COH04S1-vaccinated individuals develop robust orthopoxvirus-specific humoral and cellular responses, including cross-reactive antibodies to MPXV-specific virion proteins as well as MPXV cross-neutralizing antibodies in 45% of the subjects. In addition, NHP vaccinated with COH04S1 or sMVA show similar MPXV cross-reactive antibody responses. Moreover, MPXV cross-reactive humoral responses elicited by COH04S1 are comparable to those measured in JYNNEOS-vaccinated subjects. Finally, we show that mice vaccinated with COH04S1 or sMVA are protected from lung infection following challenge with MPXV clade IIb.1.

**Conclusions** These results demonstrate the capacity of sMVA vaccines to elicit cross-reactive and protective orthopox-specific immunity against MPXV, suggesting that COH04S1 and sMVA could be developed as bivalent or monovalent mpox vaccine alternatives against MPXV.

## Plain language summary

Mpox is an ilness caused by the mpox virus (MPXV) that belongs to the poxvirus family. The 2022-2023 mpox outbreak highlights the need to develop effective vaccines against MPXV. We have developed a COVID-19 vaccine using as scaffold chemically synthesized genetic material of a highly attenuated and safe poxvirus vector. This scaffold is the same present in a vaccine that has been approved and is given to prevent mpox. Here, we show that healthy human volunteers or monkeys vaccinated with this COVID-19 vaccine generated a robust immune response against MPXV, similar to that generated by the mpox vaccine with the same scaffold. This COVID-19 vaccine is also able to protect mice from infection caused by the MPXV strain isolated from the recent mpox outbreak. This COVID-19 vaccine in a poxvirus scaffold might be an additional tool to curtail mpox outbreaks.

[1] Department of Hematology and HCT and Hematologic Malignancies Research Institute, City of Hope National Medical Center, Duarte, CA, USA. [2] Center for Gene Therapy, City of Hope National Medical Center, Duarte, CA, USA. [3] Bioqual, Rockville, MD, USA. [4] Center for Infectious Disease and Vaccine Research, La Jolla Institute for Immunology (LJI), La Jolla, CA, USA. ✉email: flavia.chiuppesi@gmail.com

Mpox, formerly known as monkeypox, has been the center of a global health emergency that was declared from summer 2022 to spring 2023. This unprecedented mpox outbreak with epicenter in Europe and the US disproportionately impacted members of the LGBTQ+ community and racial and ethnic minority groups and caused alone in the US over 30,000 cases and 40 deaths, before the national emergency was ended on January 31, 2023. The outbreak was caused by a Clade IIb.1 mpox virus (MPXV) that demonstrated unusually efficient human-to-human transmission and that appeared to have lower case fatality rate than previously known circulating MPXV clades, I and IIa, which primarily affect Central and Western African countries, respectively[1,2]. Despite the low mortality in healthy individuals, mpox has been commonly associated with debilitating lesions and severe complications, especially in immunocompromised individuals[3].

JYNNEOS is the FDA-approved third-generation smallpox/mpox vaccine based on modified vaccinia Ankara (MVA), a highly attenuated, replication-defective orthopoxvirus vector that has been safely administered at the end of the smallpox eradication campaign to more than 120,000 individuals, including immunodeficient children and HIV-infected individuals[4]. MVA has been developed as safer alternative to smallpox vaccines based on replication-competent vaccinia (VACV) vectors, such as ACAM2000 (Dryvax), which use has been limited due to occurrence of severe side effects[5]. JYNNEOS received FDA approval as a smallpox/mpox vaccine based on a Phase 3 non-inferiority immunogenicity trial versus ACAM2000[6] as well as MPXV challenge studies in non-human primates (NHP)[7]. Due to vaccine shortage during the recent mpox outbreak, JYNNEOS received emergency use authorization (EUA) for intradermal (ID) administration, which allowed use of one-fifth of the dose normally used for the previously approved subcutaneous (SC) route of administration. Despite the proven efficacy of MVA to protect against mpox, with real-world vaccine efficacy estimates ranging between 66% and 86%[8,9], preclinical data for the capacity of MVA to elicit MPXV cross-reactive immune responses is limited, and recent clinical data has shown low-to-absent MPXV cross-reactive neutralizing antibody (NAb) responses in JYNNEOS vaccinated subjects born after the end of the smallpox vaccination campaign[10,11].

We have developed a fully synthetic MVA (sMVA) platform to reconstitute virus that is virtually identical to wild-type MVA in terms of replication properties, host cell tropism, and immunogenicity[12]. Using this platform, we have developed COH04S1, a multiantigen COVID-19 vaccine encoding for spike and nucleocapsid antigens. COH04S1 has been extensively tested in animal models, including NHP, demonstrating potent immunogenicity and protective efficacy against SARS-CoV-2 ancestral virus and emerging variants of concern[12–15]. COH04S1 has been safely tested in a Phase I, randomized, placebo-controlled clinical trial in healthy adults, resulting in stimulation of robust and durable humoral and cellular immune responses to both spike and nucleocapsid antigens[16,17]. COH04S1 is clinically the most advanced MVA-based COVID-19 vaccine, and it is currently being evaluated in three Phase 2 clinical trials (NCT04639466, NCT04977024, NCT05672355).

Here we demonstrate that COH04S1 vaccination of healthy adults elicits robust orthopoxviral-specific humoral and cellular immunity, including cross-reactive binding antibodies to MPXV virion proteins and MPXV cross-NAb responses. Additionally, we show that NHP vaccinated with COH04S1 or its sMVA vector backbone develop comparable orthopoxvirus-specific and MPXV cross-reactive humoral responses. Moreover, MPXV cross-reactive humoral responses elicited in COH04S1-vaccinated healthy adults are comparable to those measured in healthy adults vaccinated with the FDA-approved mpox vaccine JYNNEOS. Finally, we demonstrate that vaccination of MPXV-susceptible wild-derived castaneous (CAST)/EiJ mice with COH04S1 or its sMVA vector backbone results in a significant reduction of lung viral loads following intranasal (IN) challenge with MPXV Clade IIb.1. These results suggest that COH04S1 and sMVA could be developed as bivalent or monovalent mpox vaccines to expand the number of vaccine alternatives to enhance outbreak preparedness and response plans against MPXV.

## Methods

**Human subjects.** COH04S1 safety and immunogenicity was investigated at City of Hope (COH) as part of a clinical protocol (IRB#20447) approved by an external Institutional Review Board (Advarra IRB). The open-label and randomized, placebo controlled, phase 1 clinical study is registered (NCT04639466). All subjects gave informed consent at enrollment. COH04S1-induced SARS-CoV-2 immunity in this population has been described[16,17]. Here we evaluated orthopoxviral cross-reactive immune responses in samples from a subset of volunteers enrolled in this phase 1 clinical study that consented to secondary research studies. Out of the 51 subjects who received one or two doses of COH04S1, 5 subjects were selected from each dose group, for a total of 20 subjects, based on a 2-dose regimen and availability of frozen peripheral blood mononuclear cells (PBMC) samples. Subjects were not required to provide their smallpox vaccination status, and poxvirus serostatus at enrollment was not evaluated. However, an exclusion criterion was any poxvirus-vaccination within six months of enrollment in the trial. Subjects were vaccinated intramuscularly (IM) with $1 \times 10^7$ pfu (DL1), $1 \times 10^8$ pfu (DL2), or $2.5 \times 10^8$ pfu (DL3) of COH04S1. Summary of study subjects, vaccination schedule and age at enrollment is presented in Supplementary Table 1. Two subjects -one in DL1/DL1 group and one in DL2/DL2 group- were born before 1972 and therefore may have been previously vaccinated against smallpox. Plasma samples from JYNNEOS vaccinated volunteers were collected following local IRB approval (VD-259) at the La Jolla Institute for Immunology after informed consent was obtained for the evaluation of MPXV immunity. 19 volunteers vaccinated with two doses of JYNNEOS were enrolled between October and February 2023. Median time from full vaccination to sample collection was 98 days. Vaccination was via homologous (SC or ID), or heterologous (SC/ID or ID/SC) route. Volunteers' characteristics are presented in Supplementary Table 2.

**Non-human primates.** In life portion of NHP study was carried out at Bioqual Inc. (Rockville, MD). The study was conducted in compliance with local, state, and federal regulations and was approved by Bioqual (protocol 20–120) and City of Hope (protocol 20075) Institutional Animal Care and Use Committees (IACUC). African green monkeys (Chlorocebus aethiops) of unknown age were randomized by weight and sex to vaccine and control groups. NHP were vaccinated twice four weeks apart IM with $2.5 \times 10^8$ pfu of COH04S1 ($n = 6$, 5 females and 1 male weighting 2.91–7.23 kg) or sMVA ($n = 3$, females weighting 3.31–3.63 kg) diluted in PBS. Mock-vaccinated NHP immunized twice with PBS were used as controls ($n = 3$, females weighting 3.02–4.76 kg). The evaluation of SARS-CoV-2 immunity following NHP-vaccination with COH04S1 has been previously described[13].

**CAST/EiJ mice.** In life portion of the CAST/EiJ mouse study was carried out at Bioqual Inc. (Rockville, MD). The mouse study was conducted in compliance with local, state, and federal regulations and was approved by Bioqual (protocol 23-010) and City of Hope

(protocol 23014) IACUC. Mice were housed in small ventilated microisolator cages, and food and water were provided ad libitum. Sixty-four male CAST/EiJ mice, 6–8 weeks of age, were randomly assigned to groups (9 mice/group in experimental groups and 10 mice in the control group) and vaccinated twice in 4 weeks interval with either $1 \times 10^6$, $1 \times 10^7$, or $1 \times 10^8$ pfu/ml of sMVA or COH04S1 vaccines via intramuscular route. Treatments were labeled and animals were tagged to avoid confounders. Post-prime and post-boost blood samples were collected for the assessment of MPXV cross-reactive NAb titers. One mouse died post-prime vaccination and one post-boost. Death was determined to be not-vaccine related. One-month post-boost, mice were anesthetized with isoflurane and intranasally challenged with $6.03 \times 10^6$ pfu/ml of MPXV USA/MA001/2022 (expanded in house from NR-58622, BEI Resources). Weight was recorded daily for 5 days. On day 5 post-challenge lungs were collected for viral load assessment. Humane endpoint was determined as >20% weigh loss or any signs of acute distress.

**MVA and MPXV IgG Endpoint ELISA.** MVA-specific or MPXV cross-reactive binding antibodies were evaluated by ELISA. ELISA plates (3361, Corning) were coated overnight at 4 °C with 1 µg/mL of MVA expressing Venus fluorescent marker (MVA-Venus)[18], or with 1 µg/mL of B6R (40902-V08H), A35 (40886-V08H), M1R (40904-V07H) or H3 (40893-V08H1) MPXV antigens (SinoBiological) in PBS pH 7.4. Plates were washed (0.1% Tween-20/PBS), then blocked with 250 µl/well of assay buffer (0.5% casein/154 mM NaCl/10 mM Tris-HCl/0.1% Tween-20 [pH 7.6]/8% Normal goat serum [NGS]; 4% NGS for NHP samples ELISA) for 2 h 37 °C. After washing, serially diluted heat-inactivated serum in blocking buffer was added to the plates in duplicate wells. Plates were wrapped in foil and incubated 2 h at 37 °C after which plates were washed and 1:3,000 dilution of anti-human IgG HRP secondary antibody (BioRad 204005), or 1:10,000 anti-monkey IgG(H + L) HRP secondary antibody (Thermo Fisher PA1-84631) in assay buffer was added for 1 hour at room temperature. Plates were washed and developed with 1 Step TMB-Ultra (Thermo Fisher 34029). The reaction was stopped with 1 M $H_2SO_4$ and 450 nm absorbance was immediately quantified on FilterMax F3 (Molecular Devices). Endpoint titers were calculated as the highest dilution to have an absorbance >0.1 nm for all assays, with the exception of B6R (>0.3 nm), and M1R (>0.2 nm). Seroconversion was defined as a two or more times increase in baseline titer[6].

**MVA neutralization assay.** ARPE-19 cells were seeded in 96-well plates ($1.5 \times 10^4$ cells/well). The following day, 2-fold serial dilutions of serum starting from 1:10 were incubated for 2 h with MVA-Venus (multiplicity of infection [MOI] = 2). The serum–virus mixture was added to the cells in duplicate wells and incubated for 24 h. After the 24 h incubation period, the cells were imaged using Leica DMi8 inverted microscope. Pictures from each well were processed using Image-Pro Premier (v9.2; Media Cybernetics) and fluorescent cells corresponding to infection events were counted. The neutralization titer for each dilution was calculated as follows: NT = [1 − (fluorescent cells with immune sera/fluorescent cells without immune sera)] × 100. The titers that gave 50% neutralization (NT50) were calculated by determining the linear slope of the graph plotting NT versus serum dilution by using the next higher and lower NT using Office Excel (v2019). Seroconversion was defined as an increase of two or more times the baseline titer[6].

**Quantification of vaccine induced MVA-specific T cells.** PBMC were isolated from fresh blood using Ficoll and counted using

Luna-FL cell counter (Logos Biosystems). Frozen PBMCs were thawed, counted and $1 \times 10^6$ PBMCs were stimulated with MVA-Venus (MOI = 1) for 24 h in a total volume of 200 µl of RPMI media with 5% of human serum in a 96 wells plate. Unstimulated cells and PHA (20 µg/ml) were used as negative and positive controls, respectively. Anti-CD107a-APC, Golgi Plug (Brefeldin A) and Golgi Stop (Monesin) were added 4 h before staining. Cells were washed with PBS and stained 15 min at room temperature with Live and dead near IR, anti-CD3-FITC (Biolegend 300440, 1:40), anti-CD4-BV421 (BD 566703, 1:50), anti-CD8-BV605 (Biolegend 301040, 1:100), anti-CD69-PE (BD 555531, 1:10), anti-CCR7-PE/Dazzle 594 (Biolegend 353236, 1:20) and anti-CD45RA-PerCP (Biolegend 304062, 1:20). After washing, cells were permeabilized with Fix/Perm (BD) for 20 min at 4 °C. Cells were washed with Perm/Wash (BD) and intracellular stained with anti-IFNγ-PECy7 for 30 minutes at 4 °C, washed and resuspended in FACS buffer until acquisition. Cells were acquired in Attune NxT cytometer (Thermofisher) and data was analyzed with Flow Jo X software following the gating strategy described in Supplementary Fig. 2. Only two out of five DL3 volunteers had available PBMC samples for the analysis.

**MPXV plaque reduction neutralization test (PRNT).** The day before the assay, $2.5 \times 10^5$ Vero E6 cells (ATCC, CRL-1586) were seeded in each well of a 24-well plate using plate-seeding media (DMEM/Glutamax/10%FBS/Penicillin/Streptomycin). The following day, four-fold serum dilutions were prepared in 96-well plates using infection media (DMEM/Glutamax/2%FBS) and equal volumes of diluted MPXV virus stock (MA-104/US-2003, clade 2a) were added to each well. Plates were incubated for 15–18 h at 2–8 °C. After this time the serum/MPXV mix was added to Vero E6 cells and incubated 37 °C. After 1 h of gentle rocking, 0.5% methylcellulose overlay medium (0.5% methylcellulose/DMEM/Glutamax/10%FBS/Penicillin/Streptomycin) was added to each well and plates were further incubated for 48 h. Once the infection step was complete, the overlay medium was removed, and a 0.4% Crystal Violet stain solution was added to each well. After removing the staining solution, plates were scanned with a flatbed scanner and manual counting of plaques performed. Number of plaques in virus control wells were used to calculate the percentage of neutralization in samples' wells. For PRNT assays performed in the presence of complement, the virus was mixed with medium with 10% of rabbit complement (Cedarlane, CL3441-S50-R) prior to adding it to the serum samples.

**Lung viral loads (TCID50).** Lungs were collected and homogenized. Supernatants of homogenized tissues were passed through a strainer and frozen until testing. Vero TMPRSS2 cells were plated at 25,000 cells/well in complete media (DMEM/10% FBS/1% Pen-Strep/ 0.01 mg/ml Puromycin). On the following day tissue homogenates were added to the wells in quadruplicates and plates were incubated 37 °C, 5.0% $CO_2$ for 5 days. Appearance of cytopathic effect was recorded and used to calculate 50% tissue culture infectious dose (TCID50).

**Statistics and reproducibility.** Statistical analysis was performed using GraphPad Prism 8.3.0. Differences across groups were evaluated using one-way or two-way ANOVA followed by Tukey's multiple comparison test and after log transformation. T cell percentages at different time-points were compared using two-sided Wilcoxon rank test. Pearson correlation coefficients and their p values were calculated for the correlative analysis. Binding IgG and NAb titers were tested using independent

replicates. T cell responses were tested in single samples. Lungs TCID50 was derived from four independent replicates.

**Reporting summary**. Further information on research design is available in the Nature Portfolio Reporting Summary linked to this article.

## Results

**COH04S1 elicits potent orthopoxviral-specific immunity in healthy adults**. Given the FDA-approval of MVA as a vaccine against smallpox/mpox, we evaluated whether COH04S1[16] and JYNNEOS-vaccinated healthy volunteers developed orthopoxvirus and MPXV-specific immunity. First, we retrospectively evaluated orthopoxviral-specific responses in a subgroup of 20 volunteers enrolled in the recent Phase 1 clinical trial aimed at testing COH04S1 safety and immunogenicity at different dose levels (DL) (NCT04639466)[16,17]. Subjects were prime-boost vaccinated with low-dose (DL1, $1 \times 10^7$ pfu), medium-dose (DL2, $1 \times 10^8$ pfu), or high-dose (DL3, $2.5 \times 10^8$ pfu) of vaccine. Of the 20 subjects vaccinated with COH04S1, 15 (5 subjects/ group) received two DL1, DL2, or DL3 vaccinations 28 days apart, and 5 received two DL1 vaccinations 56 days apart with a placebo dose at day 28 (DL1/placebo/DL1). Four placebo-vaccinated subjects enrolled in the same trial were included as controls (Fig. 1a).

MVA-specific binding antibodies and NAb were measured in COH04S1 vaccinated subjects regardless of vaccine dose (Fig. 1b). In contrast to all placebo control volunteers, all subjects vaccinated with COH04S1 showed an increase in MVA-specific IgG titers following vaccination (Fig. 1b). Consistent with a prior dose-escalation trial of wild-type MVA[19], MVA IgG titers in DL2 and DL3 subjects following prime vaccination tended to be higher than those in DL1 subjects, indicating a dose dependent response. MVA-specific IgG titers further increased in all vaccine cohorts following the second dose, resulting in similar responses and 100% seroconversion in all vaccine cohorts (Supplementary Fig. 1). MVA-specific IgG titers slowly declined over five-months post-vaccination in all vaccine cohorts, although they remained at significantly elevated levels compared to placebo independent of the dose immunization regimen.

Following prime vaccination, only a minor proportion of DL1 subjects showed elevated MVA-specific NAb titers, whereas all DL2 and DL3 subjects showed an increase in MVA-specific NAb titers, confirming a dose-dependent vaccine effect (Fig. 1c and Supplementary Fig. 1). All subjects developed robust MVA-specific NAb titers at one month after the second dose, resulting in 100% seroconversion in all vaccine cohorts. Similar to the MVA-specific IgG titers, MVA-specific NAb titers declined over six-months post-vaccination, but they remained above baseline in most subjects. In addition, comparable MVA-specific binding antibody and NAb titers were measured in samples of non-human primates (NHP) previously vaccinated with either COH04S1 or sMVA[13] (Supplementary Fig. 1).

Orthopoxvirus-specific T cells in COH04S1-vaccinated subjects were evaluated by assessing co-expression of IFNγ with CD107a or CD69 activation markers on MVA-stimulated T cells (Fig. 1d and Supplementary Fig. 2). COH04S1 vaccinees showed a significant increase in activated IFNγ-secreting CD8[+] and CD4[+] T cells from baseline to maximal levels at one month after the first dose. After the second vaccine dose MVA-specific T cell levels remained stable, and significantly elevated levels of activated T cells were measured over six months post-vaccination. Concordant with a dose-independent induction of SARS-CoV-2 antigen-specific T cells by COH04S1[16], MVA-specific T cells levels were comparable across dose levels. As previously observed

by others, post-vaccine T cell responses to orthopoxvirus antigens were largely comprised of CD8[+] T cells[20]. Phenotypic analysis revealed that at all time-points post-vaccination, CD8[+] and CD4[+] MVA-specific T cells in COH04S1 vaccinees were predominantly T effector memory (T$_{EM}$) cells, which have been associated with protection against peripheral infection with VACV[21].

**COH04S1-vaccinated subjects develop cross-reactive immunity against MPXV**. Next, we addressed whether COH04S1 stimulated MPXV cross-reactive immune responses to known protective antibody targets of intracellular mature virus (IMV) and extracellular enveloped virus (EEV), the two major virus forms mediating poxvirus transmission and dissemination[22,23]. The selected antigens have a 93.8% to 98.4% similarity with their VACV homologs[24]. Compared to placebo controls, elevated IgG titers specific for MPXV EEV proteins B6R and A35 (homologous of VACV B5R and A33R) and IMV proteins M1R and H3 (homologous of VACV L1R and H3L) were measured in healthy human subjects vaccinated with COH04S1 at one month after the second dose (Fig. 2a). B6R, A35, M1R and H3-specific IgG titers tended to be higher in subjects vaccinated at higher DL and in DL1-vaccinated subject who received the two doses two-months apart. At six months post-vaccination, MPXV cross-reactive antibodies appeared to decline, although elevated antibody titers to MPXV antigens were consistently measured in a proportion of COH04S1 vaccinees regardless of the DL used (Supplementary Fig. 3). Elevated IgG titers against all IMV- and EEV-specific MPXV proteins were also measured in NHP after one or two doses of COH04S1 or sMVA (Supplementary Fig. 3).

MPXV cross-reactive NAb were measured in a proportion of COH04S1-vaccinated subjects using a clade IIa virus by plaque reduction assay (PRNT) (Fig. 2b and Supplementary Fig. 3). Compared to baseline, after two vaccine doses, elevated MPXV cross-reactive NAb responses were measured in all vaccine cohorts regardless of dose level (Supplementary Fig. 3). Post-boost PRNT50 titers ≥10 were measured in 9/20 volunteers (Fig. 2b). MVA-specific and MPXV-cross-reactive antibody titers induced by COH04S1 correlated strongly (Supplementary Fig. 4), indicating cross-reactivity of vaccine-induced orthopoxviral-specific responses.

**COH04S1-elicited MPXV cross-reactive immunity is similar to that induced by JYNNEOS**. To compare the stimulation of MPXV cross-reactive humoral responses between COH04S1 and FDA-approved JYNNEOS, we analyzed immune plasma from a cohort of volunteers vaccinated through different routes with JYNNEOS during the recent mpox health emergency (Supplementary Table 2). Individuals were vaccinated two times with the standard subcutaneous dose ($1 \times 10^8$ pfu), or with the low-dose-sparing regimen ($2 \times 10^7$ pfu) via intradermal route that has been recently given EUA. Additionally, some individuals were prime-boost vaccinated with a combination of the two doses/immunization routes. As shown in Fig. 2c, comparable MVA-specific IgG titers and MPXV-cross reactive IgG titers specific for EEV proteins B6R and A35 and IMV protein H3 were measured between COH04S1 and JYNNEOS vaccine cohorts. Notably, MPXV cross-reactive IgG titers specific for IMV protein M1R were significantly elevated in COH04S1 vaccinees compared to JYNNEOS vaccinated subjects. In addition, comparable MPXV cross-reactive NAb responses were measured in COH04S1- and JYNNEOS vaccinees when titers were measured both in the presence and absence of exogenous complement (Fig. 2d). A minority of individuals in each cohort (2/20 in the COH04S1 cohort and 5/19

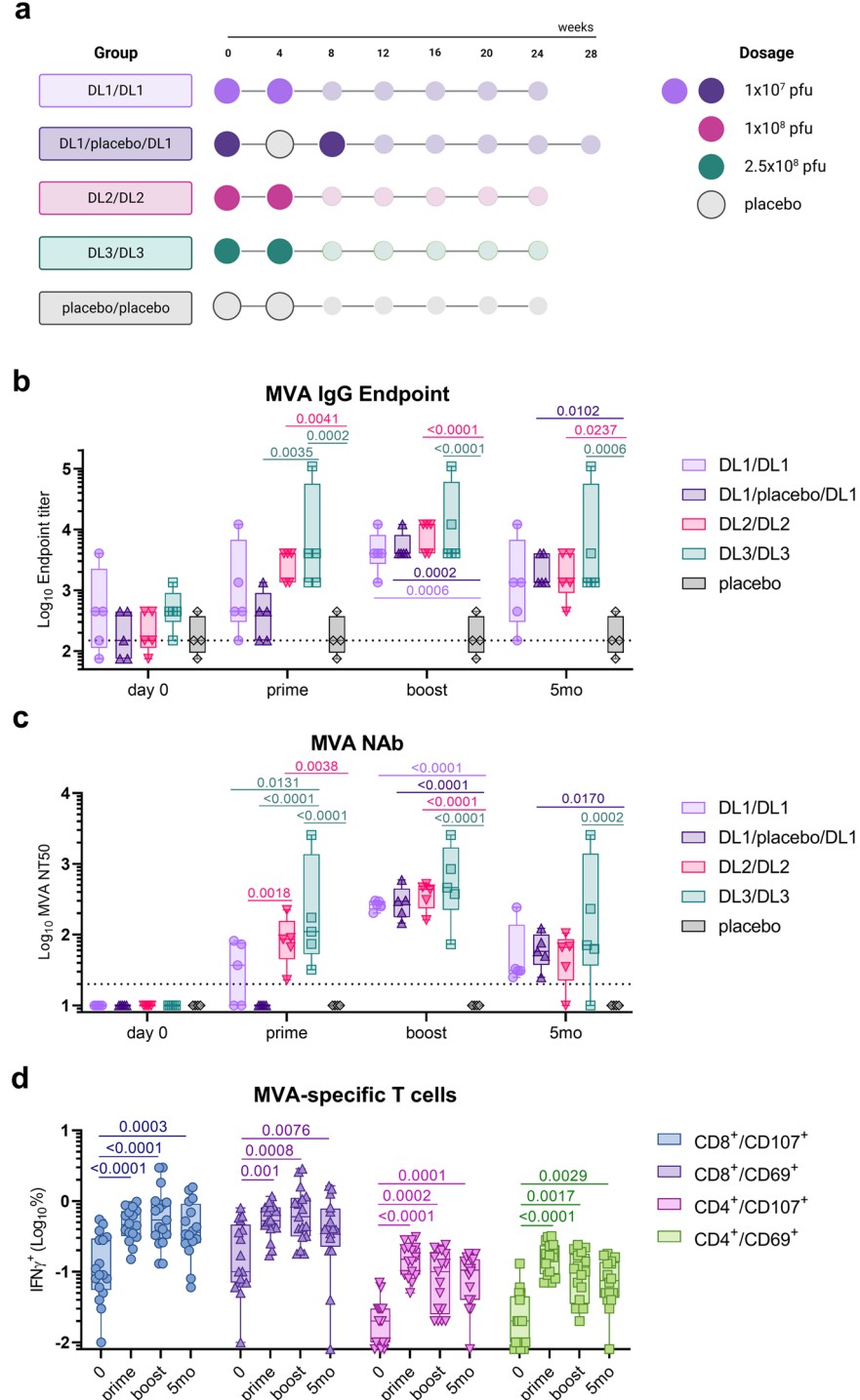

**Fig. 1 MVA-specific humoral and cellular responses in COH04S1-vaccinated healthy adults. a** COH04S1 schedule and dosing in healthy adults. Large circles indicate time of vaccination. MVA-specific humoral responses. MVA-specific IgG endpoint titers (**b**), and neutralizing antibodies (NAb) (**c**) were measured by ELISA and neutralization assay in subjects before vaccination, post-prime vaccination, and at one- and five-months post-booster vaccination with COH04S1 at dose-level (DL) 1 (DL1/DL1 [lavender circles] and DL1/placebo/DL1 [upward purple triangles]), DL2 (DL2/DL2 [downward pink triangles]), and DL3 (DL3/DL3 [green squares]) (*n* = 5 subjects/group). Placebo controls (*n* = 4 subjects [gray diamonds]) were included. **c** MVA-specific cellular responses. MVA-specific CD8+ and CD4 + T cells co-expressing CD107 (blue circles and downward pink triangles, respectively) or CD69 (upward purple triangles and green squares, respectively) markers were measured by intracellular IFNγ staining in subjects described in b-c. Box plots show 25th–75th percentiles, lines indicate medians, whiskers go from minimum to maximum values. Two-way ANOVA followed by Tukey's multiple comparison test was used in (**b**, **c**) after log transformation. Two-tailed Wilcoxon paired *T* test was used in (**d**). P values < 0.05 are shown. Dotted lines in b-c represent the lower limit of detection of the assay.

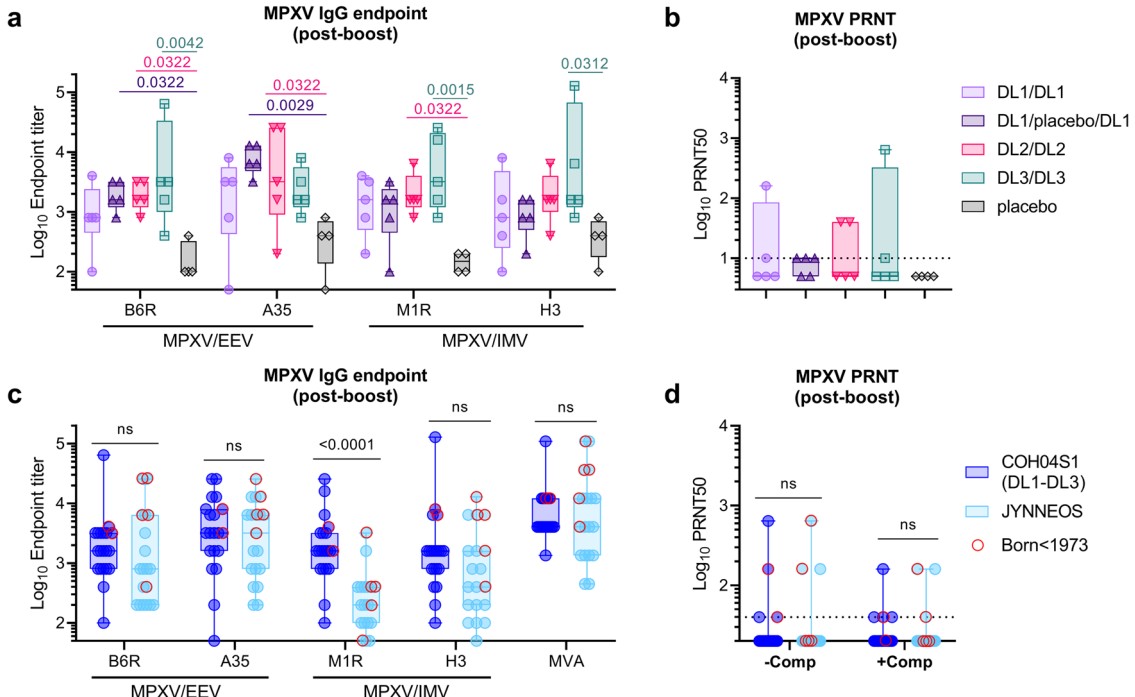

**Fig. 2 MPXV-specific humoral response in COH04S1- and JYNNEOS-vaccinated individuals.** MPXV-specific IgG endpoint titers to MPXV virion proteins B6R, A35, M1R, and H3 (**a**) and MPXV-cross reactive neutralizing antibody titers (PRNT50) (**b**) were measured by ELISA and MPXV PRNT assay in healthy adults ($n = 5$ subjects/group) one-month post-booster vaccination with COH04S1 at dose-level (DL) 1 (DL1/DL1 [lavender circles] and DL1/placebo/DL1 [upward purple triangles]), DL2 (DL2/DL2 [downward pink triangles]), and DL3 (DL3/DL3 [green squares]). Placebo controls ($n = 4$ subjects [gray diamonds]) were included. MPXV-specific IgG endpoint titers to MVA and MPXV virion proteins B6R, A35, M1R, and H3, (**c**) and MPXV-cross reactive neutralizing antibody titers (PRNT50) measured in the presence or absence of complement (**d**) were evaluated in healthy adults vaccinated with COH04S1 as described for (**a**, **b**) ($n = 20$ subjects [dark blue circles]), and healthy adults vaccinated with two doses of JYNNEOS ($n = 19$ subjects [light blue circles]). Red open circles indicate volunteers born before 1973 who likely had smallpox vaccination during childhood. Box plots show 25th–75th percentiles, lines indicate medians, whiskers go from minimum to maximum values. Two-way ANOVA followed by Tukey's multiple comparison test was used after log transformation. *P* values < 0.05 are indicated. IMV intracellular mature virions, EEV extracellular enveloped virions.

in the JYNNEOS cohort) were born before 1973 and likely received smallpox vaccination during childhood. These subjects with VACV pre-existing immunity were more likely to have elevated MPXV-cross reactive binding and neutralizing antibodies (Fig. 2c, d). These results indicate that COH04S1 vaccine elicits MPXV cross-reactive humoral responses that are comparable or even superior compared to those induced by the FDA-approved mpox vaccine JYNNEOS.

**COH04S1 and its sMVA vector backbone protect against MPXV in CAST/EiJ mice.** To evaluate the capacity of COH04S1 and sMVA to confer protection against MPXV, we assessed vaccine-efficacy against MPXV lung infection utilizing the MPXV-susceptible CAST/EiJ mouse model. CAST/EiJ mice ($n = 9$–10/group) were vaccinated twice by intramuscular route in 4 weeks interval with escalating doses of either COH04S1 or sMVA and challenged intranasally 4 weeks later with a MPXV clade IIb.1 isolate derived from a MPXV-infected individual in Massachusetts, USA in May of 2022 (Fig. 3a). As shown in Fig. 3b, MPXV-specific cross-reactive NAb were detectable in a small number of animals post-prime vaccination. MPXV-specific NAb titers significantly increased after the second vaccination in the high-dose sMVA and COH04S1 vaccine groups, while intermediate- and low-dose sMVA and COH04S1 groups showed low-to-absent MPXV-specific NAb titers after the booster vaccination.

Minimal weight gain/loss was observed after MPXV intranasal challenge, with a significant increase in body weight compared to controls only observed at days 4 or 5 post-challenge in mice

vaccinated with low-dose sMVA or intermediate- and high-dose COH04S1 (Fig. 3c). When MPXV viral load (VL) was measured in lung tissue of mice five days post-challenge, all sMVA- and COH04S1-vaccinated animals showed markedly reduced MPXV VL compared to controls, independent of the used vaccine dose (Fig. 3d). Lung VL in controls ranged from $10^7$ to $2 \times 10^9$ TCID50/g, consistent with productive lung infection by MPXV in unvaccinated animals. In contrast, mice vaccinated with low-dose sMVA and COH04S1 had lung VL that ranged from below the detection limit ($2.5 \times 10^3$ TCID50/g) to $8 \times 10^6$ TCID50/g, while mice vaccinated with sMVA and COH04S1 intermediate-dose had low-to-undetectable lung VL in most cases. Only 2/8 mice vaccinated with high-dose of sMVA and COH04S1 showed titers above the detection limit, indicating complete or potent viral control in these animals. Lung VL in sMVA and COH04S1 low-dose vaccine groups were significantly higher than in mice vaccinated with intermediate- and/or high-dose, suggesting improved protection through higher vaccine doses. Taken together, these results show that COH04S1 and its sMVA vector backbone confer protection against lung infection caused by MPXV clade IIb.1, even when administered at very low dose.

## Discussion

This study demonstrates that multiantigen sMVA-based COVID-19 vaccine COH04S1 and its sMVA vector backbone elicit robust orthopoxviral-specific responses that confer cross-reactive and protective immunity against MPXV. We show that heathy adults vaccinated with COH04S1 at different dose levels develop potent MVA-specific humoral and cellular responses as well as binding

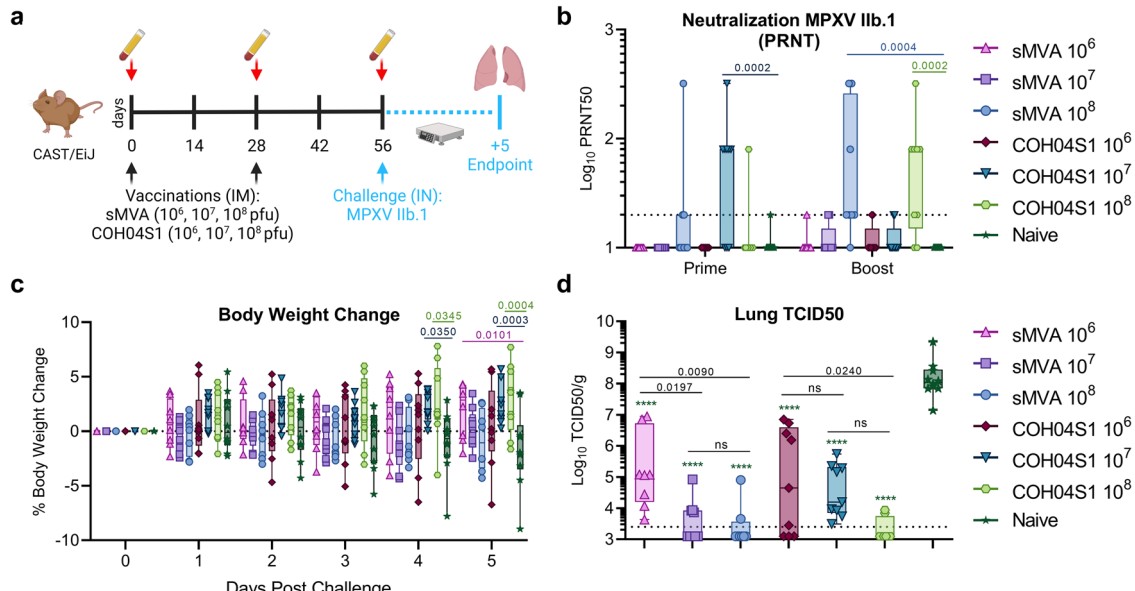

**Fig. 3 COH04S1 and sMVA immunogenicity and protective efficacy against mpox in CAST/EiJ mice. a** Study design. CAST/EiJ mice were intramuscularly (IM) vaccinated two times with sMVA at $1 \times 10^6$ ($n = 9$ mice, upward pink triangles. One mouse died post-prime due to vaccine-unrelated causes), $1 \times 10^7$ ($n = 9$ mice, purple squares), and $1 \times 10^8$ pfu ($n = 9$ mice, light blue circles. One mouse died post-boost due to vaccine-unrelated causes), or COH04S1 at $1 \times 10^6$ ($n = 9$ mice, brown diamonds), $1 \times 10^7$ ($n = 9$ mice, downward teal triangles), and $1 \times 10^8$ pfu ($n = 9$ mice, light green hexagons). Unvaccinated mice were used as controls ($n = 10$ mice [dark green stars]). Serum samples were collected at baseline, day 28 and day 56 for immunological analysis. Mice were intranasally (IN) challenged with mpox clade IIb.1 at day 56 and weight was recorded for 5 days. At day 5, lungs were collected for viral load (VL) assessment. **b** Neutralizing antibody (NAb) titers. NAb were evaluated at days 28 (post-prime) and 56 (post-boost) using a plaque reduction neutralization test (PRNT). 50% PRNT titers are shown. Dotted line indicates lower detection limit. **c** Body weight. Shown are body weight changes compared to baseline. **d** Lung VL. A 50% tissue culture infectious dose (TCID50) assay was used to evaluate lung VL at day 5 post-challenge. Dotted line indicates lower detection limit. Data are presented as box plots extending from 25th to 75th percentiles, with lines indicating medians, and whiskers going from minimum to maximum values. Two-way (**b**, **c**) or one-way (**d**) ANOVA followed by Tukey's multiple comparison test were used following log transformation. In (**d**), ****=$p < 0.0001$ when comparing each vaccine group to unvaccinated controls.

antibodies to major MPXV virion proteins and MPXV cross-reactive NAb responses. Importantly, MPXV-specific humoral responses stimulated in COH04S1-vaccinated healthy adults are similar to those measured in healthy adults vaccinated with the only FDA-approved mpox vaccine JYNNEOS, which is based on parental MVA. In addition, NHP vaccinated with either COH04S1 or sMVA develop similar orthopoxviral- and MPXV cross-reactive humoral responses, indicating comparable capacity of sMVA vectors with and without inserted antigens to elicit orthopoxvirus/MPXV-specific immunity. Finally, we demonstrate that CAST/EiJ mice vaccinated with either COH04S1 or sMVA are protected from productive lung infection following challenge with a MPXV clade IIb.1 strain of the recent unprecedented MPXV outbreak. These results suggest that COH04S1 and sMVA may represent valuable alternatives to JYNNEOS that could be used as bivalent or monovalent vaccine candidates to protect against MPXV infection.

Considering that both humoral and cellular immunity have been shown to play a protective role against orthopoxviruses[25], it is promising that COH04S1 and sMVA vaccination of healthy adults or NHP elicits robust and comprehensive orthopoxvirus-specific antibody and T cell responses. Orthopoxvirus-specific immunity induced following infection/vaccination is known to be long-lasting and to be effectively recalled by additional vaccine doses[11]. In our cohort, 2/24 volunteers (one in DL1/DL1 and one in DL2/DL2 groups) were born before 1973, and therefore may have undergone routine smallpox vaccination before the smallpox vaccination campaign was ended in 1972. Similarly, in the JYNNEOS cohort, 5/19 vaccinated subjects were born before 1973. In both cohorts those subjects with probable pre-existing immunity

appear to have above-average MVA-specific and MPXV-cross reactive humoral responses indicative of an effective anamnestic response to vaccination. This observation also suggests that additional doses beside the two that are recommended by the FDA may improve the response to vaccination. For instance, in clinical studies testing immunogenicity of MERS candidate vaccine MVA-MERS-S and Flu candidate vaccine MVA-H5, MPXV cross-reactive NAb titers went from being measurable only in a minority of volunteers after two doses to be detected in all volunteers after the third dose[10,26].

The finding that COH04S1-vaccinated healthy adults and NHP vaccinated either with COH04S1 or sMVA develop robust cross-reactive antibodies to MPXV IMV and EEV proteins suggests that sMVA-based vaccines can stimulate protective humoral responses predicted to interfere with MPXV transmission and dissemination. In addition, all COH04S1 vaccine cohorts regardless of dose developed MPXV cross-reactive NAb responses, albeit at low level, highlighting the potential of COH04S1 to elicit antibodies that are considered essential to protect against MPXV infection. The observed low levels of MPXV cross-reactive NAb titers in COH04S1 or JYNNEOS vaccinees are consistent with prior findings for JYNNEOS-vaccinated subjects[10,11,26], suggesting that other immune responses besides NAb may play a critical role for protection against mpox. While the dependency on complement of potent VACV NAb targeting EEV has been shown before[27,28], we measured comparable MPXV cross-reactive NAb both in the absence and in the presence of an exogenous source of complement in our cohorts of COH04S1 and JYNNEOS vaccinees. This contrasts with the improved MPXV cross-reactive NAb titers measured by others in JYNNEOS

vaccinees and mpox convalescents when rabbit or guinea pig complement was added during testing[11,29]. The reason for this discrepancy is not clear but it may be due to differences in the assays, in the source of complement, or in the amount of complement added during the assay[29].

Given the comparable MPXV cross-reactive immune responses observed in COH04S1 and JYNNEOS-vaccinated healthy adults, these results may suggest that COH04S1 or sMVA have the potential to confer similar efficacy against MPXV infection as recently reported for JYNNEOS[9]. In addition, a recent multijurisdictional case-control study conducted during the 2022/2023 MPXV outbreak in the United States reports an estimated JYNNEOS vaccine effectiveness against mpox of 88% in immunocompetent individuals and 70% in immunocompromised subjects[30], suggesting that COH04S1 or sMVA would be able to confer potent protection in both immunocompetent and immunocompromised individuals. Whether this estimated vaccine effectiveness of JYNNEOS against mpox applies only to protection against the less virulent MPXV clade IIb.1 or potentially also to protection against more pathogenic MPXV strains remains to be determined. This is an important question that would need to be addressed considering that MVA demonstrated potent protection against MPXV lethal virus challenge in immunocompetent macaques but failed to provide protection against lethal MPXV challenge in macaques with AIDS[31]. Intriguingly, while the measured MVA-specific titers and MPXV-cross reactive IgG titers specific for EEV proteins B6R and A35 and IMV protein H3 were comparable between COH04S1 and JYNNEOS vaccinees, the observed MPXV cross-reactive IgG titers specific for IMV protein M1R were significantly higher in COH04S1-vaccinared subjects compared to JYNNEOS vaccinees. These increased M1R-specific IgG titers observed in COH04S1-vaccinated subjects compared JYNNEOS-vaccinated health adults may suggest that COH04S1 confers improved cross-reactive immunity compared to JYNNEOS to protect against MPXV viral dissemination. While the precise reasons for the differences in M1R-specific titers between COH04S1 and JYNNEOS vaccinees remain unclear, it could be associated with the use of different vaccination routes or dose regimen, differences in the manufactured vaccine products or the viral backbone, or potentially also with the SARS-CoV-2 spike and nucleocapsid antigens incorporated in the COH04S1 vaccine.

While the efficacy of MVA to protect against MPXV infection has been demonstrated in different animal models using clade I viruses[7,32–34], the efficacy of MVA to provide protection against the less pathogenic MPXV clades IIa and IIb.1 has not been shown in preclinical models. In addition, recent mRNA vaccine approaches based on conserved MPXV-specific IMV and EEV proteins have been shown to elicit potent MPXV-specific immune responses, although vaccine efficacy was only assessed against VACV challenge, but not against MPXV[35]. A recently published study demonstrated protection by recombinant IMV and EEV proteins against MPXV clade IIb.1 in Balb/c mice[36]. Wild-derived CAST/EiJ mice have been used as an animal model of MPXV infection thanks to their higher susceptibility compared to laboratory strains[37]. We demonstrate in the CAST/EiJ mouse model that COH04S1 or sMVA confer potent protection against lung infection by MPXV clade IIb.1. This is the first challenge study in a relevant animal model showing protection by an MVA-based vaccine against the newly emerged MPXV clade IIb.1 strain. It is noteworthy that a significant reduction in lung VL in CAST/EiJ mice vaccinated with low-dose COH04S1 compared to unvaccinated controls was measured in the absence of MPXV cross-reactive NAb. This finding appears consistent with observations for JYNNEOS-vaccinated healthy adults and suggests that non-neutralizing antibody functions or cellular responses may have played a role in protecting CAST/EiJ mice

following MPXV clade IIb.1 virus challenge. Another explanation may be that the performance of the neutralization assay without complement may have resulted in overlooking complement-dependent NAb with protective functions[27,28].

These results in sum provide novel insights into the capacity of MVA-based vaccines to elicit orthopoxviral-specific and MPXV cross-reactive and protective immunity and suggest that COH04S1 or sMVA could be utilized as vaccine alternatives to protect against MPXV. The safety and immunogenicity of COH04S1 observed in the recent Phase 1 clinical trial[16,17], in addition to the strong protective capacity of COH04S1 measured in preclinical animal models[13,14], suggest that COH04S1 represent a unique vaccine candidate to simultaneously protect against SARS-CoV-2 and MPXV, which would be a desirable attribute especially in low-resource settings. The findings from this report that are highly relevant for clinical adaptation are the addition of another vaccine product that is safer than ACAM2000 to the pharamacopiea available to mpox-infected individuals.

## Data availability
All data generated or analyzed during this study are included in this published article, Supplementary Information, and Supplementary Data 1 file. All other data are available from the corresponding author on reasonable request.

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

## Acknowledgements

We would like to thank all the participants who volunteered in the study and all the investigators and study site personnel who assisted in the clinical trial completion. Funding was provided by the Carol Moss Foundation, donors Julie and Roger Baskes, Judd Malkin, Michael Sweig, and the City of Hope Integrated Drug Development Venture program. We thank Sara Gianella Weibel (Department of Medicine, University of California San Diego) for her support in finding valuable resources. We acknowledge and thank Christoph Pittius and Yuriy Shostak (Research Business Development, City of Hope) for excellent project management. We thank Christina Ulloa (Department of Hematology & HCT, City of Hope) for excellent support of investigators and meeting coordination. Figures 1a and 3a were created with Biorender.com.

## Author contributions

Study conceptualization: FC, FW, DJD. Study design: FC, FW. Investigation: FC, MG, SOF, ML, SK, MK, TK. Resources: SK, AG, AS. Manuscript writing: FC, FW, DJD. Regulatory affairs: SD. Clinical PI: JAZ. All authors contributed to and approved the final version of this manuscript.

## Competing interests

While unknown whether publication of this report will aid in receiving grants and contracts, it is possible that this publication will be of benefit to City of Hope (COH). COH had no role in the conceptualization, design, data collection, analysis, decision to publish, or preparation of the manuscript. The authors declare the following competing interests: D.J.D. and F.W. are co-inventors on a patent application covering the design and construction of the synthetic MVA platform (PCT/US2021/016247). D.J.D., F.W., and F.C. are co-inventors on a patent application covering the development of a COVID-19 vaccine (PCT/US2021/032821) and provisional applications covering MPXV. D.J.D. is a consultant for GeoVax Labs and Helocyte Inc. A.F. is a consultant for Pfizer. A.S. is a consultant for Consultant for AstraZeneca Pharmaceuticals, Calyptus Pharmaceuticals, Inc, Darwin Health, EmerVax, EUROIMMUN, F. Hoffman-La Roche Ltd, Fortress Biotech, Gilead Sciences, Gritstone Oncology, Guggenheim Securities, Moderna, Pfizer, RiverVest Venture Partners, and Turnstone Biologics. All other authors declare no competing interests. GeoVax Labs Inc. has taken a worldwide exclusive license for COH04S1 under the name of GEO-CM04S1.
