## [Peer Review File · Communications Medicine]

This manuscript has been previously reviewed at another Nature Portfolio journal. This document only contains reviewer comments and rebuttal letters for versions considered at Communications Medicine

Reviewers' comments:

Reviewer #1 (Remarks to the Author):

The manuscript by Chiuppesi F et al addressed a relevant question of the cross-reactive immunogenicity against monkeypox elicited in humans and macaques by a SARS-CoV-2 vaccine candidate (COH04S1) based on the poxvirus strain MVA. The objective was to demonstrate that this type of vaccine could elicit both SARS-CoV-2 immunogenicity and efficacy (previously shown in animal models in various publications by the group), as well as immunity against the closely related monkeypox. Using serum and PBMCs from participants in a phase I clinical trial and from a NHP study, the authors clearly documented a dose-response induction by the vaccine of monkeypox-specific immunity (binding and neutralizing antibodies, T cell responses). The authors suggested that this type of vaccine simultaneously could protect against SARS-CoV-2 and monkeypox. There are some considerations:

1. The most critical missing part of the study is the demonstration that the immune response triggered by the vaccine was effective in the control of monkeypox infection. This could be addressed using an appropriate animal model system, like the mouse CAST/EiJ, that upon challenge with 10^5 - 10^6 PFU of monkeypox induced rapid weight loss and 100% mortality by day 8 post infection (see review Hutson C.L and Damon I.K (2010). *Viruses* 2(12):2763-2776.
2. That the MVA vaccine could elicit specific immune responses against the closely related monkeypox was anticipated based on previous studies with the approved MVA vaccine from Bavarian Nordic (Jynneos). A comparison of total binding and neutralizing antibodies from individuals previously infected during the epidemic of monkeypox (should not be a problem to get serum) with those from the clinical trial will add value to the study, as indicator of extent of immunity obtained from vaccination versus infection.

Comments on the Response to Reviewer 1:

1. The response to the reviewer 1 on the comparative study on immune responses between individuals receiving the synthetic sMVA or the recombinant MVA expressing the S and N antigens of SARS-CoV-2 (COH04S1) with those immunized with the Jynneos vaccine is appropriate. The argument by the authors on differences in the route of administration of each vaccine is valid, although this data could still provide information on the strength (binding and neutralizing antibodies) for each immunogen. At least, the strength of the vaccine should have also been compared with serum from individuals infected with MPXV, available upon request in some hospitals or agencies.

In terms of the request by reviewer 1 to show efficacy of the SARS-CoV-2 (COH04S1) against MPXV as part of a preclinical study, and the response that this is out of the scope of the manuscript as this is a short note, is weak. The manuscript will benefit greatly with such efficacy data, using a well known mouse model.

2. The response on the lack of effect of the vaccine on immunosuppressed individuals is appropriate.

Reviewer #2 (Remarks to the Author):

In this study, the authors have examined crossreactive immune responses to monkeypox that are elicited by vaccination with recombinant MVA vectors. They found cross-reactive antibody titers by ELISA, measured MVA-specific T cells, and measured monkeypox-reactive neutralizing antibody responses, many of which appear to be at or below the limits of detection (1:10).

Comments

Line 81. The phrase, "...only a minor proportion..." is vague. Please provide a specific number (and percentage) of each group that showed MVA-specific neutralizing titers that increased above a specified threshold. Also

Line 87. The phrase indicating that antibody responses, "...remained above baseline in most subjects." is vague. It appears that 2 or more subjects became seronegative by month 5. Please provide the specific number of seropositive subjects (and % seropositive) for each cohort at month 5.

Lines 120-122. In the orthopoxvirus field, a plaque-reduction neutralizing titer-50 (PRNT50) of <1:10 is considered seronegative. This means that any serum sample in which less than 50% neutralization occurred at a 1:10 dilution is actually seronegative. As Fig. 2b is currently drawn, the data gives the impression that there are more seropositive subjects because they are being compared to only 4 unvaccinated controls that happen to have low background NT50 titers. The authors should provide the NT50 titers of the baseline serum samples for comparison.

The authors should state in the results what % of vaccinated subjects demonstrated at least a 4-fold increase in baseline neutralizing titer.

Looking at the methods, it appears that monkeypox neutralizing titers were performed using a 4-fold dilution series with data acquired at a 1:10 and 1:40 dilution. Please provide the % of subjects who had neutralizing titers (defined the same way as the MVA titers in Fig. 1, which is to specify the serum dilution in which at least 50% of infectious virus is neutralized). Since the neutralizing titers appear to be quite low, I would ask that the authors specifically state the % of vaccinated subjects in each group that neutralized >50% of monkeypox virus at a dilution of 1:10 and importantly, provide the % of vaccinated subjects in each group that neutralized >50% of monkeypox virus at a dilution of 1:40.

If less than half of vaccinated subjects mounted an NT50 titer of 1:40 after two vaccinations, then this represents a very weak immune response (similar to other studies) and I would ask that the title of the paper be modified to read, "Synthetic modified vaccinia Ankara vaccines confer weak cross-reactive monkeypox immunity" to make this point immediately clear to the reader and non-specialist. Also, if my assumption is correct and fewer than half of vaccinated subjects mounted a monkeypox-reactive NT50 titer of at least 1:40, then I would caution against using the term "robust" when describing the antiviral antibody responses in the abstract and results sections since this would not be the case under these circumstances.

Line 132. Although T cells may play a role in protection against orthopoxviruses in mice, an excellent study by Edghill-Smith et al. (Nature Medicine, 2005;11:740) shows that vaccine-induced memory T cells do not play a measurable role in protection against monkeypox infection of non-human primates. The authors should cite this paper and mention that antibodies are both necessary and

sufficient for protection against monkeypox but that the role of T cells in protection against monkeypox in NHP (or humans) has not been established.

Line 172. The authors mention that two subjects were born prior to 1972 and may have been vaccinated against smallpox. Can the authors state what the baseline neutralizing titers were for these two individuals since this would indicate whether they may have indeed been vaccinated? If these two subjects did indeed have higher baseline neutralizing titers indicative of prior vaccination, what happens to the % seroconversion against monkeypox at a 1:10 or 1:40 dilution? In other words, if these two subjects had pre-existing neutralizing titers, the authors should also state somewhere in the paper the % of vaccinated subjects who mounted cross-reactive neutralizing titers to monkeypox after first removing these two potential confounding subjects from the analysis.

Fig. 1.

Please clarify in the figure legend that prime represents day 28, boost represents day 56 after primary vaccination so that the reader knows the relative spacing/intervals of the samples without having to search for this information in the results.

Fig. 1b. The dashed line is set at 200 but there is no description of what this line represents. Typically, a dashed line is used to show the cutoff below which non-specific binding from naïve/unvaccinated subjects may occur. Based on the day 0 samples, I would ask that the dashed line be moved from 200 to approximately 2,000 since this would mainly segregate the naïve/day 0 baseline values below this threshold and make it easier for the reader to gauge the levels of vaccine-induced antibody responses that are above the non-specific binding observed at baseline prior to vaccination.

Fig. 1c. Please describe what the dashed line in this panel represents in the figure legend.

Fig. 1d. The non-specific MVA-reactive T cell responses are very high at baseline (day 0). This makes it difficult to establish what proportion of vaccinated participants elicited a detectable T cell response above their individual baseline. Please provide an additional panel (panel e) with bar graphs showing the percentage of subjects in each group who mounted MVA-specific T cell responses that are at least 4-fold over their baseline frequencies at each time point.

Fig. 2. Please provide the baseline (day 0) ELISA titers to match the data provided in Fig. 1.

Fig. 2a. Once the naïve/unvaccinated samples have been graphed alongside the day 56/one-month post-booster ELISA samples, please add a dashed line in which 95% (i.e., 19/20) of the naïve serum samples are below that threshold so that the reader has a guide for distinguishing between non-specific ELISA binding antibodies and vaccine-induced antibody levels.

Fig. 2b. The authors prepared Fig. 2a using the same conventions, graphing style, and scale as Fig. 1a. This is good because it helps the reader compare the data across both MVA and monkeypox. I respectfully request that the authors do the same thing for Fig. 2b and present the actual NT50 titers – many of which will be below a 1:10 (i.e., lacking at least 50% neutralization at a dilution of 1:10). The authors should use the same scale and graphing approach that they used for presenting neutralizing titers to MVA in Fig. 1c. This is important because on a scale of 10 to 10E4, it will be more obvious to the reader that the neutralizing titers to monkeypox are likely to be vastly suboptimal compared to MVA.

Supplemental Fig. 3. Please re-graph the data to follow the format used for presenting neutralizing titers in Fig. 1b and Fig. 1c and please add the baseline serum titers for comparison to the post-vaccination serum titers. Likewise, please present the NHP NT50 titers using the same approach and scale as that used for Fig. 1C. In the main text, please describe the % of animals that seroconverted ($\geq 1:10$) and the % of animals that had reached an NT50 of at least 1:40. This will help the reader to understand how immunogenic the crossreactive monkeypox antibody titers are in the NHP model in comparison to their human counterparts.

Further review of Author replies to Reviewer #2 comments:

I think that the authors responded reasonably well to the questions raised by Reviewer #2 although with the one exception that I would add when Reviewer #2 brought up a good point (Q3, vaccination history of the subjects) that the authors did not address. The authors stated that prior immunization status was not available. However, in the absence of this medical history, I would ask that they respond to Reviewer #2 by simply providing the baseline serology data that they already have in hand. The authors state that there are 2 people who were born prior to 1972 and that these two people may or may not have been vaccinated. If these people had been vaccinated, then they would continue to have vaccinia-specific antibody responses that would score positive for neutralizing antibodies in the baseline sera. The authors should provide details/comment/supplemental information for MVA-vaccinated subjects who were Seronegative at baseline (in other words this will strengthen their argument that their MVA vaccine still elicits anti-MPV antibodies in people who never had prior smallpox vaccination?). This not only would rule out the potential caveat that the 2 older subjects born before 1972 might have skewed the results in favor of cross-reactive antibodies to MPV due to childhood smallpox vaccination, but it would also rule out the other possibilities – for instance, the US military vaccinated against smallpox up until 1990 and so anyone who served in the US military during that period of time could also have pre-existing immunity from traditional smallpox vaccination. Again, this can be easily ruled out by simply providing a dataset somewhere in the paper indicating the results that are from only the people who were seronegative at baseline.

Reviewers' Comments:

We thank the Reviewers for providing excellent critical comments. We hope that we have adequately addressed all concerns raised by the Reviewers as described below.

Reviewer #1:

Remarks to the Author:

The authors capitalized on a phase I Trial in 20 human volunteers, that tested three different doses of a synthetic MVA platform expressing SARS-CoV2 antigens (COH04S1), to investigate orthopox-Monkeypox virus (MPXV) elicited immunity. In parallel they analyzed also orthopox-MPXV immune responses elicited by the COH04S1 at the highest dose (DL3/DL3) in Non-human Primate (NHP). Not surprisingly they found that that COH04S1 as well as the synthetic MVA (sMVA) induced in both species, binding antibodies to two proteins from MPXV Extracellular Envelope Virus and two from Intracellular Mature Virus, MVA specific T-cell responses (with CD8+ cells prevalence) and neutralizing antibodies to MPXV. They propose that the COH04S1 vaccine could protect against SARS-CoV2 and Monkeypox. The immunogenicity results presented here are encouraging and expected but the ability of COH04S1 to protect from monkeypox is unproven. At minimum a comparison of the MPXV neutralization titers induced by Jynneos and sMVA or COH04S1 in humans should be performed (using the same assays) as well as a preclinical study to test the ability of the sMVA to protect against lethal monkeypox. These data are essential to support the claims put forward in this paper.

We agree with the Reviewer on the importance of adding comparability and protection data. We have now included new data (Figure 2) for comparing MPXV cross-reactive binding and neutralizing antibody responses measured in COH04S1 and JYNNEOS vaccinees using the same assays, demonstrating that COH04S1 elicits similar or even superior humoral responses compared to JYNNEOS. While the measured MPXV-cross reactive IgG titers specific for EEV proteins B6R and A35 and IMV protein H3 were comparable between COH04S1 and JYNNEOS vaccinees, the observed MPXV cross-reactive IgG titers specific for IMV protein M1R were significantly higher in COH04S1-vaccinated subjects compared to JYNNEOS vaccinees. In addition, we measured comparable, albeit low levels of MPXV cross-reactive neutralizing responses between COH04S1- and JYNNEOS-vaccinated individuals. These findings are included in the results section in lines 146-164 and further discussed in the manuscript.

We also added new data (Figure 3) demonstrating the capacity of COH04S1 and its sMVA vector backbone without inserted antigens to protect MPXV-susceptible CAST/EIJ mice against productive lung infection following challenge with a MPXV clade IIb.1 viral isolate, whereby protection was observed even at low vaccine dose of 1×10^6 PFU. These data provide first evidence in a preclinical model that MVA-based vaccines can protect against MPXV of the recent 2022/2023 mpox outbreak. These findings are included in the results section in lines 166-191 and further discussed in the manuscript.

In addition, it needs to be discussed that neither sMVA nor COH04S1 are likely to protect immune compromised individuals, the most vulnerable population for MPXV. Modified vaccinia virus Ankara (MVA) has been shown to be safe in immune-compromised macaques, but is unable, even when used prior to the replication competent ACAM2000 (Dryvax) to protect against lethal MPXV despite the induction of higher neutralizing antibodies titers by the MVA/Dryvax combination. See paper "Smallpox Vaccine Does

Not Protect Macaques with AIDS from a Lethal Monkeypox Virus Challenge (Y. Edghill Smith, Mike Bray, et al) and attachment. Dryvax alone, in contrast, completely protects against MPXV healthy NHP.

We agree that this is an important point to discuss. We added to the discussion (lines 242-251): “A recent multijurisdictional case-control study conducted during the 2022/2023 MPXV outbreak in the United States reports an estimated JYNNEOS vaccine effectiveness against mpox of 88% in immunocompetent individuals and 70% in immunocompromised subjects, suggesting that COH04S1 or sMVA would be able to confer potent protection in both immunocompetent and immunocompromised individuals. Whether this estimated vaccine effectiveness of JYNNEOS against mpox applies only to protection against the less virulent MPXV clade IIb.1 or potentially also to protection against more pathogenic MPXV strains remains to be determined. This is an important question that would need to be addressed considering that MVA demonstrated potent protection against MPXV lethal virus challenge in immunocompetent macaques but failed to provide protection against lethal MPXV challenge in macaques with AIDS. An important point regarding ACAM2000 is that its use is associated with frequent development of pericarditis. In fact the US government was so concerned about the frequent development of this severe side effects that it did not advocate use of ACAM2000 for prophylaxis of mpox, even when there was a pronounced shortage of Jynneos which led to its being rationed and an alternative route of subdermal administration was advocated. Additionally administration of ACAM2000 to subjects with congenital or acquired immune deficiency disorders, including people living with HIV (regardless of immune status) is contraindicated by the CDC (<https://www.cdc.gov/poxvirus/mpox/interim-considerations/acam2000-vaccine.html>).

Reviewer #2:

Remarks to the Author:

In the manuscript by Chiuppesi et al., the authors explore and re-analyze the cross-reactive and neutralizing immunity against MPXV in a cohort of healthy adults and NHP originally prime-boost vaccinated against SARS-CoV-2 using the synthetic sMVA-based vaccine COH04S1 expressing S and N antigens at different dosage. Additionally, NHP were vaccinated with sMVA lacking S and N antigens. MVA-specific immune responses were determined by ELISA, neutralization assay and ICS; cross-reactive humoral immunity towards MPXV by ELISA and PRNT. They demonstrate MVA-specific as well as MPXV-cross-reactive immunity in the study.

The work is in principle sound and original and the data are interesting, but the study has several strong limitations.

1. There is no indication from the literature that an MVA vector expressing a clinically relevant recombinant antigen would not induce potent immune responses against its vector backbone. In contrast, measuring the anti-viral response is often used as vaccination control, particularly when immune responses against the target antigens are low or unpredictable. Therefore, it was to be expected that MVA-specific as well as MPXV-reactive responses were induced.

We agree with the reviewer that the stimulation of orthopoxviral-specific immunity by COH04S1 was to be expected considering its synthetic MVA vector backbone, although data for the stimulation of MPXV cross-reactive immunity by parental MVA is still limited, and the capacity of COH04S1 or sMVA to stimulate

either orthopoxviral specific or MPXV cross-reactive responses was uncharacterized. We therefore believe that our data for induction of MVA-specific and MPXV cross-reactive immunity by COH04S1 in vaccinated healthy adults provide new insights into capacity of MVA or sMVA vaccine vectors to elicit protective humoral responses that are considered important for protection against orthopoxviral or MPXV dissemination and transmission. In addition, new data (Figure 2) has been added for comparing MPXV-cross reactive binding and neutralizing responses elicited in COH04S1 and JYNNEOS vaccinees, revealing similar MPXV cross-reactive responses between these two vaccine cohorts. Intriguingly, while the measured MVA-specific titers and MPXV-cross reactive IgG titers specific for EEV proteins B6R and A35 and IMV protein H3 were comparable between COH04S1 and JYNNEOS vaccinees, the observed MPXV cross-reactive IgG titers specific for IMV protein M1R were significantly higher in COH04S1-vaccinated subjects compared to JYNNEOS vaccinees. These increased M1R-specific IgG titers observed in COH04S1-vaccinated subjects compared JYNNEOS-vaccinated health adults may suggest that COH04S1 confers improved cross-reactive immunity compared to JYNNEOS to protect against MPXV viral dissemination. These findings are further discussed in lines 252-263.

2. There are no human data included in the study from subjects having received sMVA. Comparative vaccine studies have been performed in NHP, but only 3 animals received sMVA and statistics are lacking. So, the important question how COH04S1 performs in humans in comparison to sMVA is not addressed, although the last phrase in the Summary implies exactly this. Therefore, a cohort of human subjects vaccinated with sMVA must be included.

Focus of this study was to characterize orthopoxviral and MPXV-cross reactive immunity elicited by sMVA-based COVID-19 vaccine COH04S1 in a recent Phase 1 clinical trial in healthy adults to assess the capacity of its sMVA vector backbone to elicit such responses in humans. A comparative study between COH04S1 and empty sMVA vector would require initiating a completely new Phase I clinical trial. The NHP data was included to provide evidence in a relevant animal model for the compatibility of COH04S1 and its sMVA vector backbone when used without inserted antigens to stimulate orthopoxviral and MPXV cross-reactive immunity, but we completely understand the limitations of the findings as they cannot substitute for human data. Because of the low number of animals in the vaccine and control groups, and differences in the number of animals between groups, statistical analysis was not initially performed. We now include the statistics for all NHP-related results (Figures S1 and S3). In addition, and as mentioned in response to point one, new data has been added demonstrating that COH04S1 elicits similar or even superior humoral MPXV cross-reactive immune responses compared to JYNNEOS vaccinated individuals.

3. The main interest of this study is the determination of MVA- specific and MPXV-cross-reactive immunity. Therefore it is mandatory to know the vaccination status of the human subjects. It may have been enough to exclude subjects with poxvirus vaccination within 6 month for the enrollment in the clinical trial investigating SARS-CoV-2-specific immunity, but it is certainly not sufficient for studying poxvirus immunity. The vaccination status must be clear as it has been shown that immunological memory to VACV lasts for decades.

We agree with the Reviewer' comment. Unfortunately, as indicated in the methods section (lines 302-303) this information on the volunteer VACV/MVA immunization status was not available. However, routine smallpox vaccination was ended in 1972, and in our cohort of 24 volunteers only two volunteers

(one in DL1/DL1 and one in DL2/DL2 groups) were older than 48 years. Similarly, in the cohort of JYNNEOS vaccinated volunteers 5/19 individuals were born before 1973. Additionally, vaccination against smallpox for work-related risk is a rare event. Therefore, we believe that the MPXV cross-reactive immunity observed in most of the volunteers is a primary response to MVA and not a recall response to smallpox vaccination. We have added a paragraph commenting on this (lines 149-155). Furthermore, in Figure 2c-d we have marked with red circles cross-reactive immune responses to MPXV in those individuals in the two cohorts born before 1973 to provide evidence of increased cross-reactive responses in subjects possibly vaccinated with VACV compared to naïve subjects.

4. There is a general confusion about specificity and cross-reactivity. MPXV-specific immunity and MPXV-specific neutralizing Abs cannot be induced by MVA, particularly as no defined epitopes for T cells or Abs are investigated in this study. Thus, immunity regarding MPXV including T cells and Abs is cross-reactive and possibly cross-neutralizing in the NT. The MPXV-specific ELISA using 4 MPXV proteins as antigens is measuring MVA-specific and MPXV-cross-reactive Abs, but not MPXV-specific Abs as stated in the Summary.

To address the issue raised by the Reviewer, we have now substituted “MPXV-specific” with “MPXV cross-reactive” throughout the manuscript and in the title.

Minor Points

1. Head line: Change to modified vaccinia virus Ankara ...

The title was modified from “Synthetic modified vaccinia Ankara confers potent monkeypox immunity” to “Synthetic modified vaccinia Ankara vaccines confer cross-reactive and protective immunity against mpox virus”.

2. Head line: Change to monkeypox cross-reactive immunity ...

The title was modified from “Synthetic modified vaccinia Ankara confers potent monkeypox immunity” to “Synthetic modified vaccinia Ankara vaccines confer cross-reactive and protective immunity against mpox virus”.

3. Line 24: It has not been tested if the same Abs which are reactive in this ELISA are able to prevent infection.

We did not claim that ELISA-reactive antibodies are neutralizing, but we indicated that both types of antibodies are induced after vaccination with COH04S1.

4. Please indicate the route of vaccination for humans and NHP in Methods

Routes of vaccinations for humans and NHP are now included in the Methods section.

5. Line 138: Change multiple to several or four.

Multiple was changed to several.

6. Line 141: Typo. One the other hand....

Typo was corrected.

7. Line 183: Comment on the homology of these 4 antigens between MPXV and MVA. At best one could compare the sera against MVA- and MPXV-derived corresponding antigens.

The selected antigens have a 93.8% to 98.4% similarity with their VACV homologues (Faraz Ahmed et al., 2022). This comment has been added to line 127 of the revised manuscript.

8. Fig.1c: Indicate prime/boost vaccination scheme and dosage.

Prime/boost vaccination schedule and dosage was included as Figure 1a.

9. Suppl.Fig.1: Statistical analysis is lacking.

Statistical analysis was not performed because only 3 NHP were vaccinated with sMVA and mock-vaccinated. We have now included log-transformed 2-way ANOVA statistical comparison of the different vaccine groups.

10. Suppl.Fig.2b/c: Statistical analysis is lacking.

Statistical analysis was performed but no significant differences were found, likely due to the group sizes. The use of 2-way ANOVA following log transformation is now clearly indicated in Figure S2 legend.

11. Suppl.Fig2b: The day 0 values for CD8+ TC (background) are quite high as compared to the post vaccination values for CD8+ TC as well as the day 0 values for CD4+ TC. How can this be explained?

The Reviewer correctly noticed high day 0 activated CD8+ T cells in COH04S1 vaccinated subjects. Similar elevated levels of pre-vaccination CD8+ T cells against MPXV were also observed by others (Grifoni et al., 2022. 10.1101/2022.09.06.506534) using a similar assay. The reason for the higher background level of activated MVA-specific CD8+ T cells compared to CD4+ T cells is not clear. One possibility is that, given MVA large genome, there may be cross reactivity of some MVA-derived peptides with antigens from viruses commonly infecting humans. Independent of that, the major result was the significant increase in MVA-reactive CD8+ and CD4+ T cells post-COH04S1 vaccination.

12. Suppl.Fig.3c: Statistical analysis is lacking.

We have now included 2-way ANOVA statistical comparison of the different vaccine groups.

REVIEWERS' COMMENTS:

Reviewer #1 (Remarks to the Author):

As previously requested, a main result provided now in the revised manuscript is the demonstration that in susceptible CAST/EiJ mice there is a dose response immunogenicity and efficacy against mpox virus following vaccination with sMVA and COH04S1 and i.n challenge with MPXV clade IIb.1. This is an important finding that strength the MVA vaccine capacity to protect against mpox virus.

Reviewer #2 (Remarks to the Author):

The authors have largely responded to my previous concerns.

The abstract states that there was a "robust orthopoxvirus-specific humoral and cellular response" - which gives the impression that this is a "robust" response to MPXV. To clarify this for the readers, I would ask that the abstract also state the % of vaccinated subjects who demonstrated neutralizing antibody responses to MPXV of >1:10 NT50 so that there is no question to the reader as to how robust the cross-reactive antibody response is to MPXV

REVIEWERS' COMMENTS:

Reviewer #1 (Remarks to the Author):

As previously requested, a main result provided now in the revised manuscript is the demonstration that in susceptible CAST/EiJ mice there is a dose response immunogenicity and efficacy against mpox virus following vaccination with sMVA and COH04S1 and i.n challenge with MPXV clade IIb.1. This is an important finding that strength the MVA vaccine capacity to protect against mpox virus.

We thank the Reviewer for their positive comment.

Reviewer #2 (Remarks to the Author):

The authors have largely responded to my previous concerns.

The abstract states that there was a "robust orthopoxvirus-specific humoral and cellular response" - which gives the impression that this is a "robust" response to MPXV. To clarify this for the readers, I would ask that the abstract also state the % of vaccinated subjects who demonstrated neutralizing antibody responses to MPXV of $>1:10$ NT50 so that there is no question to the reader as to how robust the cross-reactive antibody response is to MPXV.

We have modified the abstract to clearly state the percentage of vaccinated subjects with MPXV cross-reactive NAb titers ≥ 10 . The abstract now reads: "COH04S1-vaccinated individuals develop robust orthopoxvirus-specific humoral and cellular responses, including cross-reactive antibodies to MPXV-specific virion proteins as well as MPXV cross-neutralizing antibodies in 45% of the subjects."